# Subtelomeric *5-enolpyruvylshikimate-3-phosphate synthase* copy number variation confers glyphosate resistance in *Eleusine indica*

Chun Zhang [1,4], Nicholas A. Johnson [2,4], Nathan Hall[2], Xingshan Tian [1,5] ✉, Qin Yu [3,5] ✉ & Eric L. Patterson [2,5] ✉

Genomic structural variation (SV) has profound effects on organismal evolution; often serving as a source of novel genetic variation. Gene copy number variation (CNV), one type of SV, has repeatedly been associated with adaptive evolution in eukaryotes, especially with environmental stress. Resistance to the widely used herbicide, glyphosate, has evolved through target-site CNV in many weedy plant species, including the economically important grass, *Eleusine indica* (goosegrass); however, the origin and mechanism of these CNVs remain elusive in many weed species due to limited genetic and genomic resources. To study this CNV in goosegrass, we present high-quality reference genomes for glyphosate-susceptible and -resistant goosegrass lines and fine-assembles of the duplication of glyphosate's target site gene *5-enolpyruvylshikimate-3-phosphate synthase* (*EPSPS*). We reveal a unique rearrangement of *EPSPS* involving chromosome subtelomeres. This discovery adds to the limited knowledge of the importance of subtelomeres as genetic variation generators and provides another unique example for herbicide resistance evolution.

*Eleusine indica* (goosegrass) is one of the most economically important weed species in tropical and sub-tropical regions globally, commonly infesting cereals (especially rice), legumes, cotton, vegetable crops, and turf systems. Decades of using the herbicide glyphosate for goosegrass control has applied enormous selective pressure for glyphosate resistance evolution. Glyphosate is a non-selective herbicide that targets the protein 5-enolpyruvylshikimate-3-phosphate synthase (EPSPS)[1], an essential enzyme for aromatic amino acid synthesis in plants. In some cases, glyphosate resistance in goosegrass is caused by EPSPS target site mutations, such as the Pro106 single mutations and the Thr102Ile and Pro106Ser (i.e., TIPS) double mutation[2]; however, increased *EPSPS* copy number variation (CNV)[3], a type of genomic structural variation (SV), is the more frequent mechanism[3] in this species. With both mechanisms present in goosegrass, some populations, or even individuals, can have both *EPSPS* CNV and target-site mutations[4], with some, most, or all the copies of *EPSPS* having one or multiple mutations.

Genomic SV can have profound effects on an organism's evolution[5]. As opposed to other forms of genetic variation, such as single nucleotide polymorphisms (SNPs), SV does not always occur at a

[1]Guangdong Provincial Key Laboratory of High Technology for Plant Protection, Institute of Plant Protection, Guangdong Academy of Agricultural Sciences, Guangzhou, Guangdong, P.R. China. [2]Department of Plant, Soil, and Microbial Sciences, Michigan State University, East Lansing, MI, USA. [3]Australian Herbicide Resistance Initiative (AHRI), School of Agriculture and Environment, University of Western Australia (UWA), Perth, Australia. [4]These authors contributed equally: Chun Zhang, Nicholas A. Johnson. [7]These authors jointly supervised this work: Xingshan Tian, Qin Yu, Eric L. Patterson. ✉e-mail: 1070470768@qq.com; qin.yu@uwa.edu.au; patte543@msu.edu

constant rate[6]. Instead, SV formation is punctuated by several factors including the environment, transposable element activity, genetics, hybridization events, and chromosome organization[7]. In plants, SV is a broad category and may include smaller-scale events like trans/cis-duplications, tandem duplications, and inversions as well as large scale events like chromosome arm inversions and even polyploidy[8]. SV that varies gene copy number (i.e., CNV) can have a direct impact on gene expression without changing the nucleotide sequence of the gene itself. Furthermore, additional gene copies often diverge over time and can eventually neo- or sub-functionalize, resulting in increased genetic diversity[9].

Some regions of the genome, as well as some gene families, are especially prone to generating SVs and CNVs. Chief among these are regions of highly repetitive sequences where unequal crossing over happens frequently due to misalignment of sister chromatids, homologous chromosomes, and even non-homologous chromosomes[10]. Chromosomes are often highly repetitive at their ends in the telomeres and subtelomeres. The subtelomeres are loosely defined and vary across taxa but are typically the next 50–100 kb of genome adjacent to the telomeres. Subtelomeres, while generally gene poor, can be sources of novel CNV events as crossing over can happen frequently. For instance, it has been previously shown in *Phaseolus vulgaris* that certain pathogen resistance genes exist in or near the subtelomere and that due to their proximity to the subtelomeres, these R genes are highly duplicated leading to certain pathogen resistance phenotypes[11]. This phenomenon is also relevant in monocotyledonous species. In allohexaploid bread wheat (*Triticum aestivum*), there is generally less synteny of genic regions among subgenomes in the subtelomeres compared to interstitial regions between the centromeres and subtelomeres, partially from high levels of CNV[12]. CNVs in subtelomeres are not limited to plants; in human subtelomeres, CNVs constitute around 80% of the most distal 100 kb of the chromosomes[13].

In plants, we are only beginning to understand the importance of SV as a novel source of genetic variation due to the advent of ubiquitous and inexpensive genome sequencing technologies[14]. Projects like the 1001 Genomes Project[15] are thoroughly investigating SV and its importance in *Arabidopsis thaliana*. In addition, there have been many other examples demonstrating the importance of SVs in the evolution of crops and non-crops and the massive effects that they can have on phenotypes[6,16–18]. One of the most striking examples of SV in action has been the evolution of glyphosate (Roundup) resistance in weedy species that effect row crop production. In cases where CNVs cause glyphosate resistance, the plant over-produces the EPSPS enzyme, to a degree that an enormous amount of glyphosate is needed to inhibit the entire EPSPS protein pool.

At least nine divergent, monocot and eudicot weed species have independently evolved glyphosate resistance via *EPSPS* CNV; an astounding example of convergent evolution[19]. Furthermore, each species has evolved these CNVs uniquely. The first *EPSPS* CNV was discovered in *Amaranthus palmeri* (Palmer amaranth)[20]. It was eventually discovered that the *EPSPS* gene is being duplicated by a novel, extra-chromosomal, circular piece of DNA named "the replicon"[21,22]. The replicon independently replicates from *A. palmeri*'s core genome and tethers itself to the core genome during cellular division[22,23]. Other weed species have duplicated *EPSPS* in more familiar ways, for example, *EPSPS* is duplicated in *Bassia scoparia* (kochia) in tandem and is thought to be the result of a combination transposable element activity and unequal crossing over[24,25]. Efforts to identify mechanisms of CNV formation have been concentrated on eudicot species in the Americas, although globally, weedy monocot species are more problematic.

Despite *E. indica*'s global economic importance, molecular biology and genomics research has remained difficult due to the lack of a quality reference genome and other molecular tools[26,27]. In this work, we generate a nearly end-to-end assembly of a glyphosate-susceptible (GS) goosegrass individual and a near complete assembly of a glyphosate-resistant (GR) individual. Furthermore, we use genomic resequencing, comparative genomics, and transcriptomics to identify the complete genomic region involved in goosegrasses *EPSPS* CNV event and provide insight into the mechanism driving increased gene copy numbers. This work is an in-depth investigation into the nature of *EPSPS* CNV of this grass species and describes how subtelomeric-repeats can drive herbicide resistance.

## Results and discussion

### Genome assembly, annotation, and overview

We assembled a chromosome-scale genome of a GS *E. indica* plant using PacBio Sequel II and long-range interaction (Hi-C) datasets. The assembly consists of nine chromosomes spanning 509,878,742 base pairs estimated to be ~97.8% complete by Benchmarking Universal Single-Copy Orthologs (BUSCO)[28] analysis with an LTR Assembly Index (LAI)[29] score of 18.77 (Supplementary Tables 1 and 2). Gene models for this species were called using a de novo Isoseq dataset and predicted genes were prescribed function using homology and other protein domain predictive software. Ultimately 27,487 gene models were predicted in the GS *E. indica* genome (BUSCO: 92.1%; Supplementary Table 2). In addition, we assembled another, chromosome-level (>99% in 15 contigs), 541,164,105 base pair long genome using a separate PacBio HiFi dataset from a single GR individual (Supplementary Table 1). This GR genome was estimated to be 97.8% complete by BUSCO with an LAI score of 16.85 (Supplementary Table 2). Using the same annotation pipeline, 29,090 genes were predicted in the GR *E. indica* genome (BUSCO: 92.2%; Supplementary Table 2). This resistant individual was confirmed to have increased *EPSPS* gene copy-number as its major glyphosate resistance mechanism in previous work via qPCR of *EPSPS* and *EPSPS* sanger sequencing[4].

On average, gene density and transposable element density vary inversely in both the GS and GR genomes; gene density decreases near the centromeres but increases in the distal parts of the chromosome arms, with the opposite being true for transposable element density (Fig. 1). There are higher numbers of LTR transposons (i.e., *Copia* and *Gypsy*), as well as other transposable elements, clustering at the centromeres (Fig. 1). The *Gypsy* superfamily is especially prevalent, with the highest transposable element density (Fig. 1). The GS genome contains 59.89% total repetitive sequence, including 2.53% DNA transposons and 40.21% retro-elements (38.29% of which are long terminal repeat elements). The GR genome contains 58.30% total repetitive sequence, including 2.08% DNA transposons and 35.71% retro-elements (33.91% of which are long terminal repeat elements). The most dominant transposon family in both genomes were classified as *Gypsy* transposons (~25%) in both genome assemblies, with *Copia* transposons being the second most abundant (~11%). All other transposons make up less than 2% of each genome. There were no large differences at the superficial scale in which we annotated repeat content between GS and GR.

*E. indica* has *Arabidopsis*-type telomere sequence (TTTAGGGn)[30] that are tandemly repeated at the terminal ends of the chromosomes in up to 39,800 bp long stretches. In the GS genome, chromosomes two, three, four, and nine have these tandem telomeric repeats at one terminal end and the remaining chromosomes, one, five, six, seven, and eight do not contain any terminal tandem telomeric repeats (Fig. 2). In the GR genome, chromosomes one, two, four, seven and eight begin and end with tandem telomeric repeats, and chromosomes three, five, six, and nine have tandem telomeric repeats at only one end of the chromosome (Fig. 2). This indicates we have captured most but not all the full-length chromosomes in the GR genome. The increase in telomere-to-telomere coverage in the GR genome compared to the GS may be explained by biological factors, such as increased homozygosity of the resistant line, or computational factors, such as different amounts of input PacBio or PacBio read N50 size. Most likely the

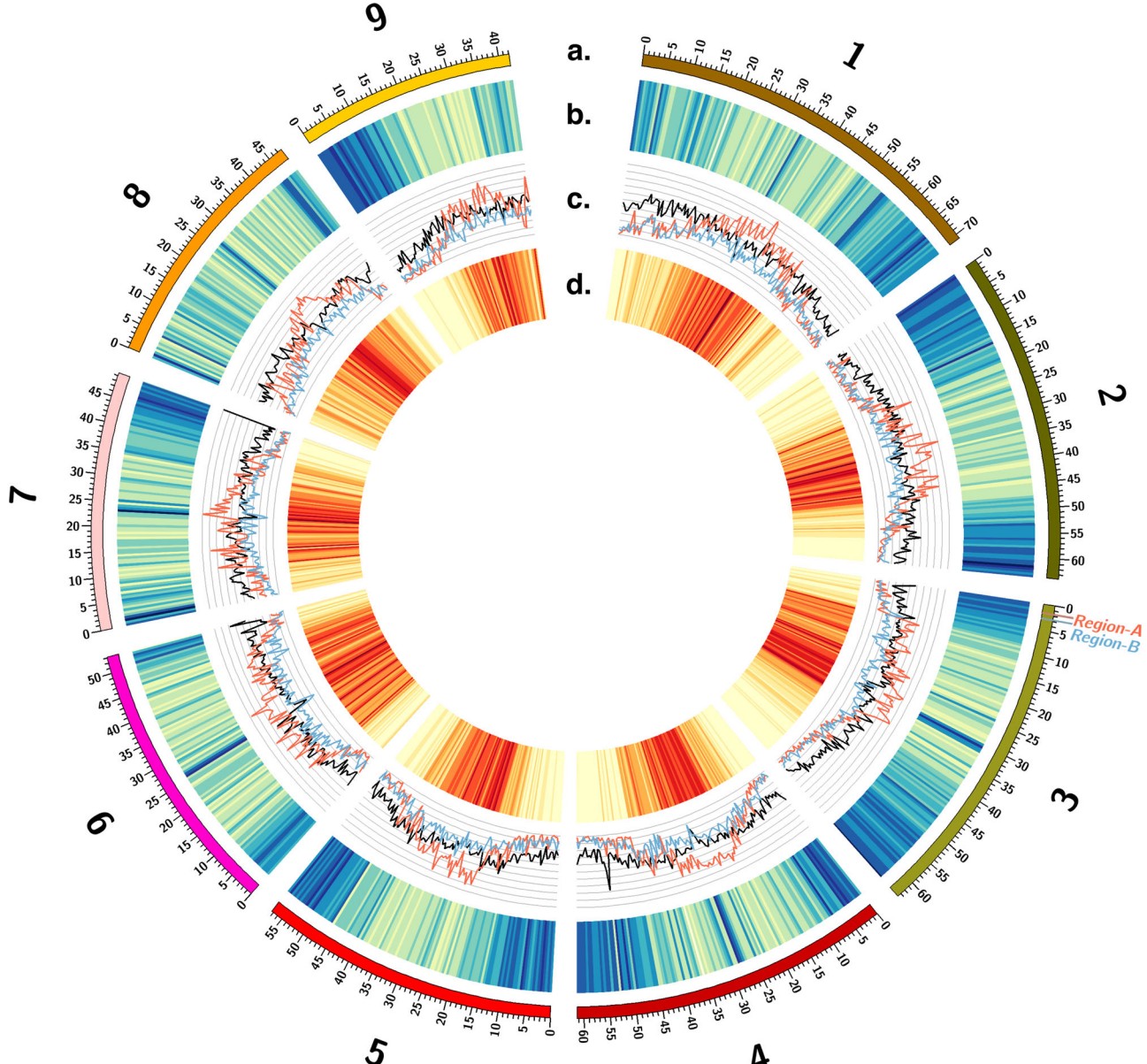

**Fig. 1 | Overview of the glyphosate-susceptible *Eleusine indica* genome.** The Circos plot shows (**a**) the length (Mb) of chromosomes one through nine as an index with corresponding (**b**) gene-rich (blue) and gene-poor (yellow) genomic regions, (**c**) *Gypsy* (red), *Copia* (blue), and other (black) transposable element family coverage across the genome (scale: 0–50%), (**d**) transposable element rich (red) and transposable element poor (yellow) genomic regions, and the native locations of Region-A (red label) and Region-B (blue label) of the subtelomeric *EPSPS*-cassette. Source data are provided as a Source Data file.

highly repetitive chromosome ends and redundancy between the telomeric repeats makes further refinement difficult without the implementation of novel techniques specifically designed to resolve these regions.

The GS and GR assemblies are highly syntenic, as indicated by the linear arrangement of large (>10 kb) lengths of nearly identical sequence that remain in order over entire chromosomes (Fig. 2). Due to the high amount of synteny between the assemblies, the slightly more fragmented GR genome has been manually ordered and named to maximize alignment with the GS genome. In the GR assembly, chromosomes three and five are composed of two contigs each while chromosomes six and seven are comprised of three contigs, with the remaining chromosomes being in single contigs.

Two large inversions were assembled in the GR genome: one at the end of chromosome five, and the other in the middle of chromosome three (Fig. 2). When reads were mapped from both GR and GS

*E. indica* PacBio datasets to the inversion junction on chromosome five of both the resistant and susceptible genome it revealed that many reads support this inversion on chromosome five in GR, while none supported the current orientation of chromosome five in GS (Supplementary Fig. 1). This may indicate an assembly error in the GS genome, however, Hi-C data from the GS genome supports its current orientation (Supplementary Fig. 2). When the same analysis was run on the inversion junctions on chromosome three, the results were inconclusive with reads mapping poorly to both the GS and GR, both upstream and downstream of the inversion (Supplementary Fig. 3). Again, Hi-C data supports the orientation of chromosome three in GS (Supplementary Fig. 2). Both inversions are flanked by complex, repeat- rich DNA which may partially explain poor read mapping. The presence of these inversions on both chromosomes three and five of GR can be confirmed using optical mapping or Hi-C on resistant lines in the future as needed, but without this additional data, we cannot say

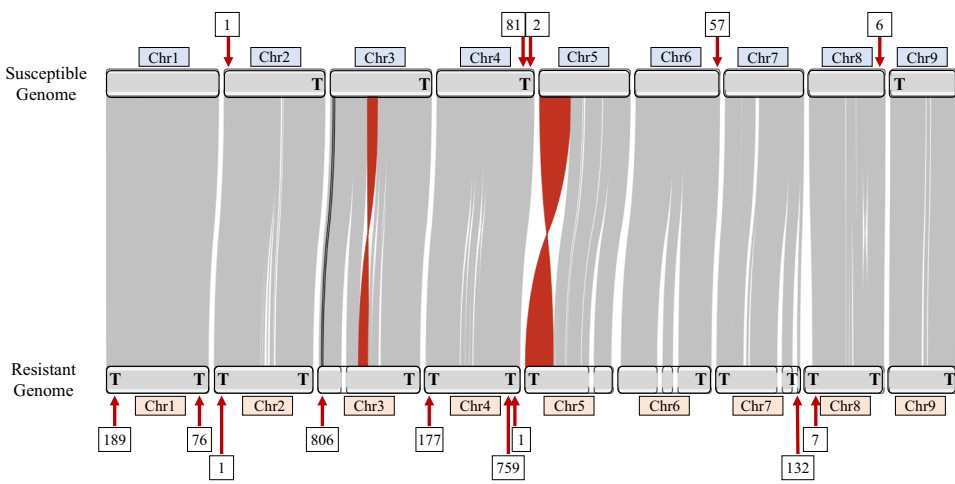

**Fig. 2 | An ideogram of glyphosate-susceptible and glyphosate-resistant *Eleusine indica* genome alignment.** Gray links indicate shared synteny between chromosome pairs. Red links indicate large inversions of synteny between the genomes. Black links represent Region-A and Region-B of the *EPSPS*-cassette in their native locations. Numbers in boxes above and below the ideogram indicate the number of copies of sub-telomeric repeats at each locus. A bold letter "T" on the karyotype represents ends of the chromosomes where the assembly reaches the telomeres. Source data are provided as a Source Data file.

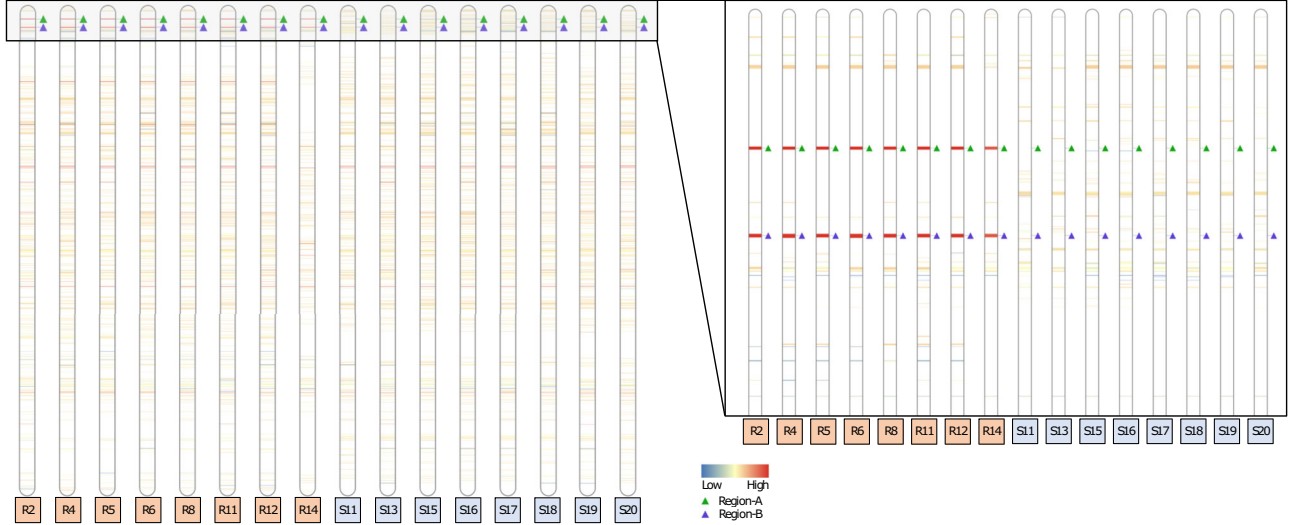

**Fig. 3 | Copy number variation in chromosome three across eight glyphosate-resistant and eight glyphosate-susceptible *Eleusine indica* individuals.** The ideogram shows deletions below 0.25× of average read depth (blue color spectrum), copy number variation above 0.25 of average read depth and below 4× of average read depth and with a *p*-value > 0.01 (white), and duplications above 4× of average read depth (red color spectrum) across chromosome three in eight glyphosate-resistant (R) versus eight glyphosate-susceptible (S) *E. indica* individuals at a scale of full chromosome length (63,742,515 base pairs) and a zoomed in view of the first 5,000,000 base pairs of chromosome three. Band thickness is proportional to the length of the genomic region exhibiting copy number variation. Region-A (green triangle), containing *EPSPS*, and Region-B (purple triangle), the genomic region co-duplicated with *EPSPS*, are consistently duplicated around 25× compared to average read depth in R individuals but are not duplicated in any of the S individuals. Source data are provided as a Source Data file.

for certain how supported these inversions are; however, given that neither impact the region containing *EPSPS* they are unlikely to be involved in glyphosate resistance.

In both the GS and GR genomes, *EPSPS*, glyphosate's target, was located ~1.6 Mb from the telomere of chromosome three. In the GR genome, *EPSPS* was also identified in 23 short un-scaffolded contigs, but always co-assembled in these un-scaffolded contigs with a copy of a sequence normally 2.4 Mb from the telomere of chromosome three, about 1 Mb downstream of *EPSPS*. We call the original location of *EPSPS* in the genome "Region-A," and the region co-assembled with Region-A, "Region-B" for ease of discussion and clarity (Figs. 1 and 3). Contrary to the tandem *EPSPS* duplications conferring glyphosate resistance in the weed *Bassia scoparia* (kochia)[24,25], only one copy of Region-A was assembled at the native location on chromosome three in the GR

genome, suggesting at least one initial trans-duplication event of the *EPSPS* must have occurred before subsequent duplications.

## *EPSPS* structural variation

Eight GS and eight GR *E. indica* genomes were re-sequenced using Illumina HiSeq X Ten to determine both the uniformity of the *EPSPS* CNV event in terms of structure but also in terms of other polymorphisms such as SNPs and InDels. This re-sequencing data was aligned to the GS genome and analyzed for (1) changes in read depth (indicating duplications or deletions) of certain regions that were uniform across the GR population and divergent from GS individuals, and (2) breaks in read alignment that describe rearrangements, translocations, inversions, and duplications. There were 34 shared predicted CNV duplication events among all GR individuals, 15 in the

pseudomolecules and 19 in the un-scaffolded contigs. However, only 2 CNVs, CNV2 and CNV3, are at the copy number predicted by copy number variation PCR. CNV2 (Region-A; average read depth: 22.03) and CNV3 (Region-B; average read depth: 22.46) had nearly identical read depth in all individuals (Table 1 and Fig. 3). Seven of the eight GR individuals had *EPSPS* read depths of ~25–29× above background with the exception being sample R14 which only had 8× (Fig. 3). From this observation we believe this sample was a heterozygote, while the other seven GR individuals had copies in both the paternal and maternal genome. In addition, these results indicate both that Region-A and Region-B are co-duplicated in every copy of the *EPSPS* CNV, and that Region-A and Region-B were translocated and fused prior to CNV proliferation. Four other CNV events (CNV21, CNV26, CNV 29, and CNV30) that were predicted to have high copy number (>20×) but are solely located in un-scaffolded contigs that are mainly composed of highly repetitive DNA and/or transposons. All GS individuals had

normal read depths for Region-A and -B and therefore no CNV in either Region-A or -B (Fig. 3).

Region-A is ~35 kb-long at the coordinates Chr3:1,666,751-1,701,750 in GS and Chr3:2,163,092-2,198,095 in GR. Region-A contains glyphosate's target, *EPSPS*, as well as four other predicted genes: A390, A400, A410, and A440. Region-B is ~41 kb-long at coordinates Chr3:2,719,751-2,760,750 in GS and Chr3:3,206,579-3,247,578 in GR with four predicted genes: B510, B520, B560, and B570 (Table 2). The TIPS double amino acid substitution of *EPSPS* was only found in sample R14. In R14, T102I was present in 12% of the 583 cDNA reads that mapped to *EPSPS* and P106S was present in 12% of 562 reads, each in approximately one out of the eight predicted copies. This indicates that as mentioned previously, this individual is heterozygous with one haplotype containing the *EPSPS* CNV and the other containing the TIPS mutation. Possibly associated with the TIPS mutation, a synonymous alanine substitution (GCA to GCG) was also found 28 amino acid residues upstream from T102I on all reads containing the TIPS mutations in the same 12% read coverage. The co-occurrence of TIPS and *EPSPS* CNV in this individual is not unexpected because it has been shown previously that *E. indica* individuals can have stacked resistance mechanisms and fitness penalties associated with mutations can be compensated for with unmutated copies of *EPSPS*. Such scenarios were predicted when the TIPS double mutation was first identified[2] and later discovered in goosegrass[4]. Furthermore, it indicated that SNPs should also be carefully looked for in GR weeds with low *EPSPS* CNV frequency as a potential primary source of glyphosate resistance.

In GR individuals, Region-A and Region-B are fused and associated with a 3396 bp region of unknown origin inserted at the beginning of Region-B, labeled 'Region-I' (Fig. 4A, B). Region-I contains no predicted genes or features of significance we can decern. Together, Region-A, Region-B, and Region-I makeup the entire 'EPSPS-cassette', a structure not found in any of the glyphosate-susceptible individuals (Fig. 3a). Flanking the *EPSPS*-cassette on both sides is a 451 bp subtelomeric sequence that can also be found in the most distal part on various chromosomes in the susceptible and resistant *E. indica* genomes as well as to the subtelomere from other grass species. On one side of the cassette the subtelomeric sequence is repeated 12 times in the forward and 31 times in the reverse directions and serves to invert the *EPSPS*-cassette. On the other side is a much larger array of repeats consisting of a minimum of 43 repeats in the forward and 294 repeats in the reverse directions (Fig. 3a). We were able to span and assemble across the small array of repeats, however the larger array became ambiguous. Given previous fluorescent in situ hybridization (FISH) experiments *EPSPS* exists on two chromosomes in GR *E. indica*[31]. This information indicates that the *EPSPS*-cassette is not scattered or widely dispersed but located in tandem on the ends of one or two chromosomes.

To verify the *EPSPS*-cassette model presented here and specifically verify each region-junction, PacBio resequencing data were aligned to the *EPSPS*-cassette (explicitly junctions checked are: Subtelomere-A, A-B, B-I, I-B, and B-Subtelomere). Around 545 PacBio reads were aligned to the manually assembled *EPSPS*-cassette model to support the Subtelomere-A and B-Subtelomere junctions, 466 reads support the A-B, B-I, and I-B junctions, and 265 reads support the subtelomere junction between the reverse and forward *EPSPS*-cassettes (Fig. 3b). The number of reads supporting the inversion is almost exactly half of the number of reads that support all other junctions, indicating that the inversion exists half as much as the others. That is to say, the full-length cassette actually consists of one forward and one reverse copy of the A-B fusion, joined by the inverted, shorter subtelomeric repeat and flanked by much larger subtelomeric sequences on both sides.

In addition, an RNA-seq experiment was performed to investigate gene expression changes driven by the *EPSPS* CNV. Although the entire *EPSPS*-cassette and all genes within it are co-duplicated in GR

**Table 1 | Duplicated CNVs shared in glyphosate-resistant (GR) individuals that do not appear in any glyphosate-susceptible individuals**

| CNV event number | Chromosome or scaffold | Start | Stop | Length | Average read depth in GR |
|---|---|---|---|---|---|
| CNV1 | Chr1 | 54,792,251 | 54,809,000 | 16,749 | 2.39 |
| CNV2 (A) | Chr3 | 1,666,751 | 1,701,750 | 34,999 | 22.03 |
| CNV3 (B) | Chr3 | 2,719,751 | 2,767,250 | 47,499 | 22.46 |
| CNV4 | Chr4 | 30,001,501 | 30,024,250 | 22,749 | 3.31 |
| CNV5 | Chr4 | 32,703,001 | 32,731,250 | 28,249 | 9.19 |
| CNV6 | Chr5 | 10,538,001 | 10,552,000 | 13,999 | 2.23 |
| CNV7 | Chr5 | 13,291,251 | 13,298,500 | 7249 | 2.22 |
| CNV8 | Chr6 | 120,001 | 124,750 | 4749 | 2.18 |
| CNV9 | Chr6 | 142,751 | 153,250 | 10,499 | 3.40 |
| CNV10 | Chr7 | 27,384,251 | 27,401,750 | 17,499 | 2.03 |
| CNV11 | Chr7 | 35,146,001 | 35,150,750 | 4749 | 2.14 |
| CNV12 | Chr7 | 40,076,251 | 40,112,750 | 36,499 | 3.68 |
| CNV13 | Chr8 | 42,437,251 | 42,510,250 | 72,999 | 2.06 |
| CNV14 | Chr9 | 11,015,251 | 11,024,750 | 9499 | 5.86 |
| CNV15 | Chr9 | 18,044,251 | 18,072,250 | 27,999 | 2.80 |
| CNV16 | Scaffold12 | 46,751 | 94,500 | 47,749 | 3.29 |
| CNV17 | Scaffold12 | 121,751 | 135,250 | 13,499 | 4.66 |
| CNV18 | Scaffold26 | 99,251 | 118,500 | 19,249 | 4.71 |
| CNV19 | Scaffold29 | 32,251 | 58,000 | 25,749 | 3.04 |
| CNV20 | Scaffold29 | 88,251 | 124,750 | 36,499 | 4.28 |
| CNV21 | Scaffold30 | 13,251 | 123,250 | 109,999 | 32.46 |
| CNV22 | Scaffold35 | 10,001 | 120,500 | 110,499 | 2.93 |
| CNV23 | Scaffold36 | 1 | 38,500 | 38,499 | 3.04 |
| CNV24 | Scaffold36 | 52,001 | 88,750 | 36,749 | 2.16 |
| CNV25 | Scaffold36 | 104,751 | 117,250 | 12,499 | 5.38 |
| CNV26 | Scaffold44 | 1 | 93,250 | 93,249 | 47.34 |
| CNV27 | Scaffold45 | 1 | 50,250 | 50,249 | 2.07 |
| CNV28 | Scaffold45 | 68,001 | 91,000 | 22,999 | 5.88 |
| CNV29 | Scaffold47 | 1251 | 80,000 | 78,749 | 45.20 |
| CNV30 | Scaffold49 | 1 | 77,750 | 77,749 | 65.48 |
| CNV31 | Scaffold51 | 1 | 23,250 | 23,249 | 2.25 |
| CNV32 | Scaffold55 | 1 | 16,250 | 16,249 | 3.84 |
| CNV33 | Scaffold56 | 37,251 | 65,500 | 28,249 | 6.15 |
| CNV34 | Scaffold58 | 48,501 | 64,000 | 15,499 | 4.42 |

(A) Region-A, containing *EPSPS*, and (B) Region-B, the region co-duplicated with Region-A.

**Table 2 | RNA-seq data of genes within the *EPSPS*-cassette**

| Gene ID | Label | Annotation | Coordinates | | | RNA-seq | |
|---|---|---|---|---|---|---|---|
| Glyphosate susceptible (GS) Glyphosate resistant (GR) | | | Start GS Start GR | Stop GS | Stop GR | log2FC | *p*-value |
| **Region-A** | | | | | | | |
| EleInSChr3g081370 | EPSPS | *EPSPS* | 1,669,076 | 1,673,225 | | 4.6 | 1.6e-11 |
| EleInRChr3_2g092340 | | | 2,165,535 | 2,169,359 | | | |
| EleInSChr3g081410 | A410 | *Ribosomal subunit protein* | 1,673,439 | 1,675,161 | | 5.8 | 2.8e-11 |
| EleInRChr3_2g092360 | | | 2,169,409 | 2,171,640 | | | |
| EleInSChr3g081390 | A390 | *tRNA 2'-phosphotransferase 1* | 1,675,833 | 1,680,441 | | 4.6 | 4.9e-13 |
| EleInRChr3_2g092350 | | | 2,172,099 | 2,176,989 | | | |
| EleInSChr3g081400[a] | A400 | Unknown protein | 1,682,414 | 1,687,866 | | 1.2 | 0.42 |
| None. | | | 2,178,757 | 2,184,208 | | | |
| EleInSChr3g081440 | A440 | Unknown protein of *E. coracana* | 1,695,565 | 1,701,288 | | 5.2 | 3e-14 |
| EleInRChr3_2g080580 | | | 2,192,031 | 2,198,135 | | | |
| **Region-B** | | | | | | | |
| EleInSChr3g082510 | B510 | *DNA repair protein RadA-like* | 2,724,808 | 2,735,415 | | 5.2 | 6.9e-11 |
| EleInRChr3_2g221600 | | | 3,211,487 | 3,224,071 | | | |
| EleInSChr3g082560 | B560 | *6-phosphofructokinase 1 (PFK) gene, complete CDS* | 2,742,394 | 2,747,873 | | 0.41 | 0.78 |
| EleInRChr3_2g119210 | | | 3,229,211 | 3,234,785 | | | |
| EleInSChr3g082520[a] | B520 | *Putative dual specificity* | 2,748,924 | 2,752,196 | | 1.1 | 0.03 |
| None. | | *protein phosphatase DSP8* | 3,235,752 | 3,239,024 | | | |
| EleInSChr3g082570[b] | B570 | *E3 ubiquitin-protein ligase 1* | 2,758,205 | 2,759,136 | | – | – |
| EleInRChr3_2g091630 | | | 3,245,033 | 3,245,964 | | | |

*P*-value of differential expression was determined using two-sided quasi-likelihood *F*-test.
[a]Not annotated in GR genome. Coordinates from BLAST of GS gene against GR genome.
[b]Filtered from differential expression plot during edgeR processing.

individuals, four of the five genes in Region-A are significantly overexpressed (*p*-value < 0.01 and fold-change>2), while only one out of four genes in Region-B is significantly overexpressed (Table 2 and Supplementary Fig 1). Genes overexpressed other than *EPSPS* from Region-A include A410: *Ribosomal Subunit protein*, A390: a *tRNA-2'phosphotransferase*, and A440: a protein of unknown function (Table 2). Interestingly, homologs of A410 and A390 are also co-duplicated with *EPSPS* in *Bassia scoparia*, a eudicot weed with a tandemly duplicated *EPSPS* CNV[24,25]. Given the annotation of these proteins, it is unlikely they are directly involved in the *EPSPS* CNV formation. B510 is the only significantly overexpressed (log fold-change: 5.2 and *p*-value: 6.9e-11) gene in Region-B (Supplementary Fig. 4 and Table 2). Gene B510 encodes a RadA-like protein, a type of protein known to be associated with *EPSPS* in glyphosate resistance in other prominent weed species such as *Bassia scoparia*. This gene involvement in the formation of the *EPSPS* CNV is unknown; however, its overexpression indicates it is currently active. RadA proteins are DNA-dependent ATPase that process DNA recombination intermediates and are therefore involved in repairing DNA breaks[32]. These proteins are particularly interesting in the case of *EPSPS* duplication due to their role in catalyzing homologous recombination. Whether or not RadA is directly involved in the duplication of the *EPSPS* loci from various weed species or merely coincidental is open for examination.

### Subtelomeres in *Eleusine indica*
Whole genome alignment reveals highly conserved subtelomeric repeat sequence from the EPSPS-cassette near the ends of many of the assembled chromosomes in both the GS and GR goosegrass genomes (Fig. 2). Interestingly, over twice as many subtelomeres at higher average copy number in each region was assembled in the GR genome (Fig. 2). There is an especially high copy number of subtelomeric repeats in chromosomes one, three, four, and seven of the GR genome that could be sites of meiotic recombination and subtelomere

rearrangement in and between chromosomes (Fig. 2)[33]. It has been previously shown in *Phaseolus vulgaris* that similar subtelomeric sequences are predisposed to unequal intra-strand homologous recombination[34] commonly resulting in large duplications of whole pathogen resistance gene pathways.

The 451bp-long subtelomeric repeat unit that flanks the *EPSPS*-cassette is most like the subtelomeric repeat region of second contig that comprises chromosome three from the GR genome (99.556%), indicating chromosome three is likely the location of the *EPSPS*-cassette in GR plants. *EPSPS* is natively on chromosome three in both assembled genomes. The subtelomeric sequence of chromosome four in both the GS and GR genomes are also highly similar to the *EPSPS*-cassette subtelomeric repeat region (98.884%). The subtelomeric repeats found on chromosomes one and seven of the GR genome and chromosome six of the GS genome are 95.778%–96.882% similar to the subtelomeric region of the *EPSPS*-cassette, while chromosomes eight and two of both the GS and GR genomes are the least related (86.301%–87.113%). The subtelomeric repeats found on chromosomes four and eight of both genomes are identical (Fig. 5; Table 3). Work by other researchers using FISH cytometry have shown that the *EPSPS* CNVs in goosegrass exist on one or possibly two chromosomes[31]. Given the sequence similarity of the subtelomeric repeat in the *EPSPS*-cassette and the subtelomeres on chromosomes three and four, translocation of the *EPSPS*-cassette between these two regions through non-homologous recombination seems feasible, however, we assembled chromosome four of the GR genome from telomere to telomere, completely through the subtelomeric region, and found no evidence of the *EPSPS*-cassette on chromosome four in this population (Fig. 2).

### Subtelomeres in plant evolution
Subtelomeres of eukaryotic organisms are hotspots for adaptive evolution due to frequent, error-prone recombination events during meiosis that lead to rapidly changing genes. In common bean

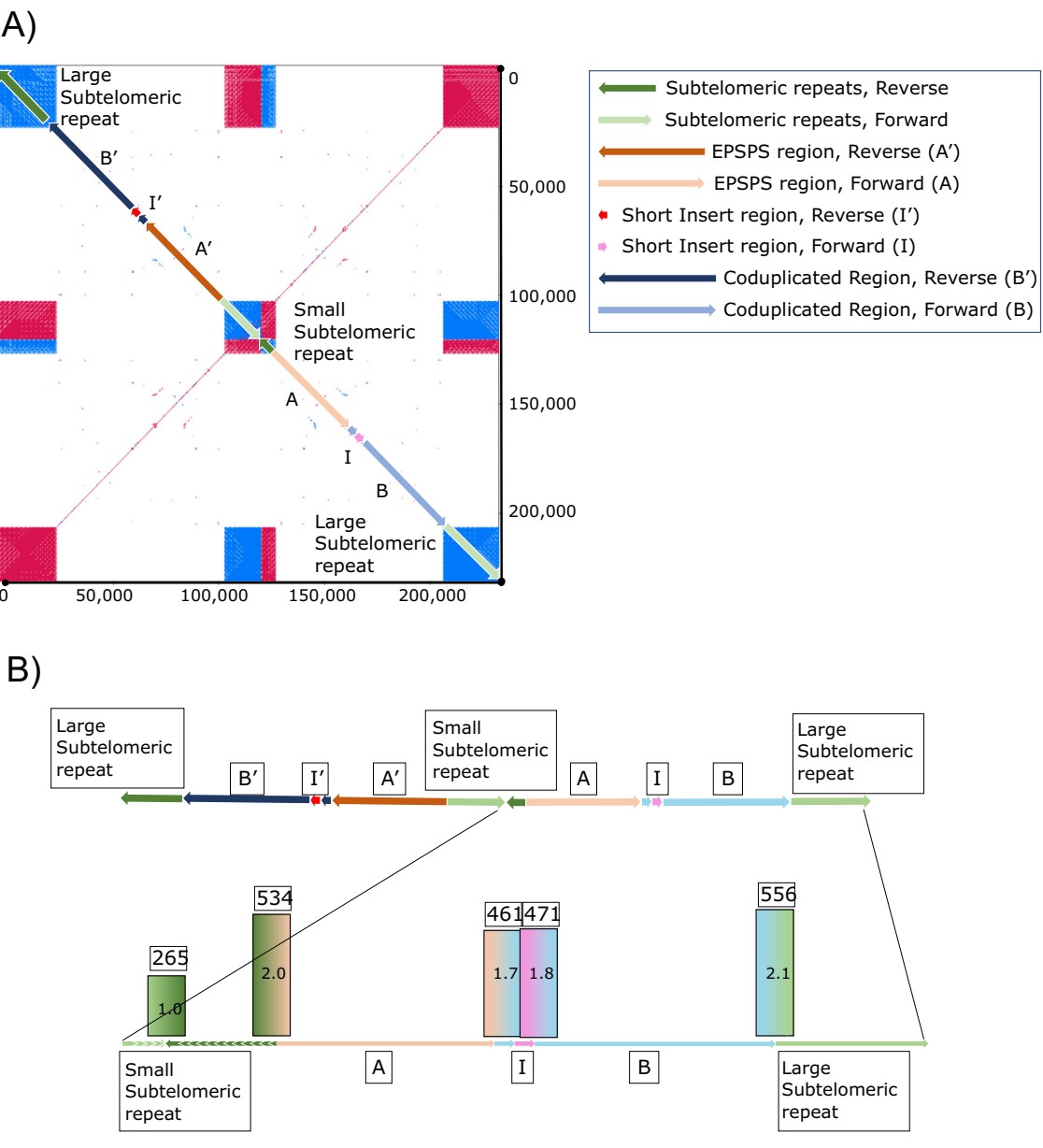

**Fig. 4 | A self-alignment of the *EPSPS*-cassette assembled from the glyphosate-resistant genome.** (**A**: the upper panel) The *EPSPS*-cassette consists of several domains. Region-A is ~35 kb and corresponds to Chr3:1666751-1701750 and contains the *EPSPS* gene itself. Region-B is ~41 kb and corresponds to Chr3: 2719751-2760750. Region-I is a small, 450 bp sequence inserted into the beginning of Region-B from an unknown origin. The entire *EPSPS*-cassette is assembled in reverse orientation, and it is denoted as A', B', and I' in reverse orientation. A shorter stretch of subtelomeric repeats (472 bp tandem repeat units) separates the forward and reverse copy of the *EPSPS*-cassette, and a larger stretch of repeats flanks the two *EPSPS*-cassettes on either end. (**B**: the lower panel) PacBio reads from the resistant genome were aligned to the forward copy of the *EPSPS*-cassette to validate the junctions of each domain (STs-A, A-B, B-I, I-B, and B-ST) and to quantify their abundance. All junctions were confirmed to be present and assembled correctly. Furthermore, we confirmed that the inversion point of the *EPSPS*-cassette in the small subtelomeric repeat region was half as abundant (set at 1x coverage/265 reads). All other junctions were shown to be approximately twice as abundant (461-556 reads). Source data are provided as a Source Data file.

(*Phaseolus vulgaris*), segmental duplications, sometimes up to 100kb-long, of disease resistance genes located in the subtelomeres are facilitated by non-homologous end joining and frequent inter-chromosomal recombination[34]. The large duplications of these distal disease resistance genes allow for divergence of homologous genes into paralogs to allow novel resistance that accounts for the rapid evolution of pathogens. Analogous to the generation of novel disease resistance genes in the subtelomeres of common bean, CNV increases variation and diversity of virulence genes in the subtelomeres of several eukaryotic pathogens including *Plasmodium falciparum*, *Trypanosoma brucei* and *cruzi*, and *Pneumocystis carinii* and sugar metabolism genes in yeast[35]. Similarly, the translocation of genomic

regions associated with herbicide or other abiotic stress resistance, like the *EPSPS*-cassette to the subtelomeric region, may allow for novel abiotic resistance to develop from the rapid accumulation of variation between duplicate loci in addition to an increase in transcript abundance.

Duplications of the *EPSPS*-cassette in the subtelomeres, especially in a grass species such as goosegrass, may not be unusual given subtelomeres' propensity for generating and selectively maintaining genetic variation in other grasses. In *Oryza sativa* (rice) the subtelomeres are theorized to be associated with high rates of transcription, recombination, and novel gene generation because they contain a large amount of highly similar paralogs, including some stress-

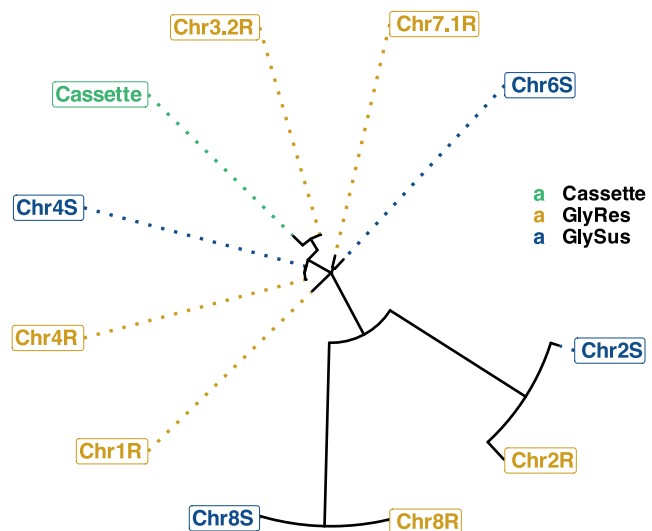

**Fig. 5 | Relatedness of *EPSPS*-cassette subtelomere sequence to chromosomal subtelomeric sequences of the glyphosate-resistant and glyphosate-susceptible *Eleusine indica* genomes.** The below plot shows the relatedness of subtelomeric sequences found on the glyphosate-resistant (R; gold) and glyphosate-susceptible (S; blue) *E. indica* genomes to the subtelomeric sequence found on the *EPSPS*-cassette (Cassette; green). Chromosomes at branch tips further from Cassette are less related to the *EPSPS*-cassette than chromosomes closer to Cassette. Branch distance is based on similarity. The sequences with the highest relatedness to the *EPSPS*-cassette subtelomere sequence on each chromosome were used as representative sequences to make this tree. Source data are provided as a Source Data file.

**Table 3 | Similarity of subtelomeric repeats throughout the glyphosate-susceptible and glyphosate-resistant *Eleusine indica* genomes to the *EPSPS*-cassette subtelomeric repeat unit**

| Subtelomeric repeat label | Percent similarity (%) |
|---|---|
| Cassette | – |
| Chr3.2 R | 99.56 |
| Chr4R | 98.88 |
| Chr4S | 98.88 |
| Chr6S | 96.88 |
| Chr7.1R | 96.44 |
| Chr1R | 95.78 |
| Chr8R | 87.11 |
| Chr8S | 86.99 |
| Chr2R | 86.3 |

This table compliments Fig. 5.

response genes[36]. Despite the apparent stochastic nature of some rice subtelomeric regions, other subtelomeric regions in *Sorghum haplensis* (sorghum), *Brachypodium distachyon* (purple false brome), and several species of the *Oryza* genus exhibit duplications of the same genomic regions and preferential gene conversions within these duplicated regions, suggesting continuous concerted evolution and selective conservation of certain genes in subtelomeric regions[37]. Such precise selectivity is also evident in several *Avena* species (oats), where a subtelomeric, 12-gene cluster developed since the evolutionary divergence of *Aveneae* is maintained with gene order colinear to the biosynthetic pathway[38]. Given the importance of subtelomeres in other monocots for generating genetic variation related to adaptive evolution and our findings, we hypothesize that *E. Indica EPSPS* is duplicated in the subtelomeres of this species after an initial translocation event. Furthermore, we speculate the *EPSPS* CNV could possibly be propagated through intra- and/or inter-chromosomal unequal crossover promoted by the flanking subtelomeric repeat regions[10] which share high similarity within and between chromosomes, especially given the high frequency of recombination within the subtelomeres[34–36] (Fig. 5; Table 3).

In summary, the goals of this research were to both obtain high quality genomic resources for *E. indica*, a major, global weed, and to use those resources to investigate the genomic rearrangements and mechanism(s) that perpetrated *EPSPS* gene duplication and therefore glyphosate resistance in this species. By assembling both a GS and GR *E. indica* chromosome-level genome, genomic resequencing eight individuals of both populations, performing RNA-seq on eight individuals from each population, and manually curating the assembly surrounding the *EPSPS* locus, we have discovered that the *EPSPS* gene in GR *E. indica* has been fused with another part of the genome, inserted in one or more of the subtelomeric regions of the genome, and duplicated an average of 25 times. We hypothesize that after the initial translocation and fusion, *EPSPS* duplication has carried on through unequal crossing over of the subtelomeres on chromosome three and

potentially other chromosomes, facilitated by the high frequency of recombination and similarity of the subtelomeric sequences in the distal chromosome ends, which future work could investigate. This work adds subtelomeric rearrangements to the list of mechanisms by which CNVs are generated and cause herbicide resistance, as well as to the relatively limited information we have about the importance of subtelomeres as genetic variation generators and hotspots for adaptive evolution.

## Methods
### *Eleusine indica* tissue generation
One glyphosate-susceptible (GS) and one glyphosate-resistant (GR) population of *E. indica* that were characterized in a previous study were collected from Guangdong Province, China[4]. GR Well-phenotyped individuals (i.e., confirmed susceptible and resistant to glyphosate) were self-pollinated for increased homozygosity and consistency of glyphosate susceptibility or resistance phenotypes. The GR population was confirmed to have *EPSPS* copy number variation by DNA quantitative PCR before purification.

Seeds of purified GS and GR biotypes were sown on wet filter paper in Petri dishes in a climate chamber at 28–30 °C, with 12 h/12 h light/dark period and 70% relative humidity. The two-leaf stage seedlings were transplanted into 28 × 54 cm trays (50 plants per tray) filled with potting soil and grown in a glasshouse. At the tillering stage, about ten individuals were randomly selected each from the GS and GR *E. indica* population and characterized. Three tillers of each plant were separated and repotted (one tiller per pot, 60 pots in total). One tiller of each plant was used for glyphosate resistance and susceptibility phenotyping, one for *EPSPS* CNV estimation and one for subsequent sequencing.

For glyphosate resistance and susceptibility phenotyping, one regrowth tiller (three days after tiller cloning) was treated with commercial glyphosate (41% glyphosate isopropylamine salt, 400 g ai ha-1 for GS and 1600 g ai ha-1 for GR), and GR (i.e., survivors) and GS (i.e., killed) phenotypes were determined three weeks after treatment. *EPSPS* CNV was again assessed in the resistant plants to ensure the CNV event was still present before genomics work began. Leaf material from untreated tillers of corresponding resistant and susceptible plants was used for genomic DNA isolation using the Plant Genomic DNA kit (Trans Gen Biotech Beijing Co., LTD). Quantitative PCR was performed using published primer pairs and methods where *EPSPS* copy number was compared to the single copy acetolactate synthase (ALS) gene as the internal reference. The untreated tiller of a confirmed GS plant was used for genomic sequencing performed at Shanghai OE Biotech Co., Ltd (Shanghai, China).

## Susceptible genome assembly

The initial GS genome was generated from 112.2 Gb (223x) of raw PacBio Sequel II sequencing data and assembled using Falcon[39] (version 0.5). The resulting initial genome was 518,672,752 base pairs long in 239 contigs with a contig N50 of 18.8 Mb. Error corrections were conducted using the Arrow[40] (version 1.0) algorithm and a hi-coverage Illumina dataset. The error corrected assembly was compared to the NCBI[41] nucleotide database using BLAST (version 2.9.0) to identify possible bacterial or mammalian contamination, of which none was found. After polishing and removing contamination from the assembly, the assembly was 519,224,895 base pairs long in 239 contigs with a contig N50 of 18.8 Mb.

To further increase the continuity of the assembly and fix any possible large-scale errors, Hi-C data was obtained. The Hi-C library utilized the Dpn II restriction enzyme (GATC cut sites) to generate sufficient digestion of the fixed DNA[42]. The final library was sequenced with Illumina HiSeq resulting in 150 base pair long paired end reads. Fastp (version 0.20; Chen et al.[34]) was used to clean the Hi-C paired end reads of adaptors and redundant PCR reads and perform analysis on the cleaned reads. Linker sequences and reads with ≥5 N (not AGCT) bases were also removed. Sliding window (window size of 4 base pairs) was performed to excise windows with an average base quality score below 20. Filtered reads less than 75 base pairs long or with an average base quality score below 15 were removed. The resulting clean Hi-C reads had a total yield (G) of 93.48, 641,488,988 read pairs, a Q20% of 97.41%, a Q30% of 92.69%, and a GC content of 44.28%.

Juicer[43] (version 1.6) was used with the default parameters of both bwa-mem (https://github.com/lh3/bwa) and 3D-DNA[44] to scaffold the contigs of the PacBio only assembly. Contigs were then clustered by contact point proximity and sorted to generate a Hi-C interaction matrix that was imported into juicebox[45] for visualization and manual inspection. The resulting matrix presented no abnormalities and contigs were able to be clustered into 9 chromosome scale scaffolds and 154 much smaller scaffolds. Gaps of 500 Ns were added between each contig to link the chromosomes for filling later. The scaffolded assembly was 519,302,895 base pairs long with an N50 of 57.27 Mb.

Finally, PBjelly[46] (version 1.0) was used to gap filling the assembly by aligning the original PacBio sequencing data to the Hi-C assembled genome. The gap filled assembly was now 522,502,607 base pairs long with a scaffold N50 of 57.37 Mb and a contig N50 of 42.10 Mb. Arrow[40] was then used for self-comparison and another round of error corrections. Last, second-generation sequencing data was used for two rounds of Pilon[47] (version 1.24) error correction, resulting in a final assembly that was 522,557,097 base pairs long with a scaffold N50 of 57.37 Mb, a contig N50 of 42.10 Mb, and 62 total gaps. The final assembly was benchmarked for gene content using BUSCO[28] (version 5.4.2) of 1375 single-copy genes from the embryonic plants database (embryophyta_odb10), and the 1348 (97.8%) were identified as either single or multi copy. Output from the Extensive de novo Transposable Element Annotator (EDTA)[48] pipeline was used to calculate an LTR Assembly Index (LAI) score of 18.77 for the GS assembly (Supplementary Table 2).

## Resistant genome assembly

The initial GR genome was generated from 28 Gb (~53×) of PacBio HiFi[49] sequencing data, assembled using HiCanu[50] (version 2.1) with a predicted genome size of 492 Mb. The resulting initial genome was 541,164,105 base pairs long in 2014 contigs with a contig N50 of 47.4 Mb. Per the instruction of HiCanu, post-assembly error corrections were not conducted to avoid introducing errors and dropping below the 99.99% accuracy rating. A BUSCO assessment against embryophyta_odb10 rated the resulting assembly as 97.8% complete (Supplementary Table 2). An LAI score of 16.85 was calculated for the GR assembly from the output of the EDTA pipeline (Supplementary Table 2). GR contigs were then aligned to the GS genome as a reference

to identify chromosomes using Minimap2 (version 2.24; citation). While most contigs were syntenic, two putative inversions were detected in GR relative to GS; the first in the arm of chromosome five, and the other in the middle of the largest contig of chromosome three. To verify these inversions were accurate and not assembler errors, 200,000 randomly selected PacBio reads from the resistant genome PacBio reads were aligned to the inversion junction points in both the GS and GR genomes. Reads spanning the junctions imply support while truncated read alignment indicates incorrect assembly (Supplementary Fig. 1 and Supplementary Fig. 3). Hi-C data from GS was used to confirm the current orientation of the GS genome (Supplementary Fig. 2).

## Genome annotation

Assembled GS and GR *E. indica* genomes were both annotated using a custom genome annotation pipeline developed by the International Weed Genomics Consortium. First, repeat regions were annotated using RepeatModeler[51] (version 2.0.2) and then masked using RepeatMasker (version 4.1.2; http://www.repeatmasker.org) and bedtools[52] (version 2.30.0) as a measure of data reduction before further annotation. IsoSeq reads were then mapped to both repeat-masked genomes using Minimap2[53] (version 2.24) to determine sites of transcription. The resulting Sequence Alignment Map (SAM) files were converted into Binary Alignment Map (BAM) files using the SAMtools[54] (version 1.11) view command before being collapsed using cDNA Cupcake (version 28.0; https://github.com/Magdoll/cDNA_Cupcake). The genomes, collapsed cDNA Cupcake outputs, repeat libraries from RepeatModeler and a protein FASTA file from a close relative, *Eleusine corocana* (Phytozome genome ID: 560), were fed into MAKER[55] (version 3.01.04) to predict the genomic coordinates of putative gene models. Genes that produced proteins under 32 amino acids long were removed from further annotation with only the longest proteins from each gene and unique untranslated regions (UTRs) used for functional annotation.

Functional annotation began by first selecting the longest isoforms from each gene using AGAT (version 0.8.0; https://github.com/NBISweden/AGAT) and gffread[56] (version 0.12.7). Longest isoforms sequence similarity searches were conducted using MMseqs2[57] (version 4.1) with NCBI, UniRef 50[58], and the InterPro[59] database using InterProScan 5[60] (version 5.47-82.0) locally. Protein localization was predicted using MultiLoc2[61] (version 1.0). Using this pipeline, 27,487 genes in GS (BUSCO: 92.1%) and 29,090 genes in GR (BUSCO: 92.2%) were predicted.

## Genome-resequencing and transcriptomics

Illumina 150 bp paired-end sequences were generated from the DNA of eight GS and eight GR individuals from the above-named populations. DNA was extracted using leaf material from untreated tillers of corresponding GS and GR plants using the Plant Genomic DNA kit (TIANGEN, Beijing, China). DNA was sequenced using the Illumina HiSeq X Ten sequencing platform (Illumina Inc., San Diego, CA, USA) with an average of 30× coverage. Illumina reads were cleaned using FASTQ and aligned to the susceptible genome using HiSat2[62] (version 2.1.0) using standard options for paired-end reads. CNVnator[63] (version 0.4.1) was used to scan the read depth from the alignments in 5 kb windows to roughly call regions of the genome that deviated significantly from the average read depth. CNVnator outputs were intersected using bedtools[52] (version 2.30.0) intersect so that regions that were amplified in all eight GR individuals but none of the GS individuals were identified. Only two such regions were discovered. The region containing *EPSPS* on chromosome three was labeled "Region-A", while the other, was labeled as "Region-B" for further analysis.

Illumina 150 bp paired-end sequences were generated from the RNA of the same eight GS and eight GR individuals that were used for genome resequencing. RNA was extracted from the leave sheath

material using the mirVana miRNA Isolation Kit (Ambion). Illumina reads were cleaned and aligned to the GS genome predicted transcriptome using HiSat2 using standard options for paired-end reads. Alignments were converted into count tables using SAMtools (version 1.11). The count table was loaded into R (version 4.2.0) and differential gene expression was calculated using package edgeR[64] (version 3.38.1) two-sided quasi-likelihood F-tests.

### Investigation of the *EPSPS*-cassette and subtelomeres

The *EPSPS*-cassette model was resolved by first using BLAST to identify all contigs containing EPSPS, Region-A, and Region-B. Contigs of interest were self-aligned in YASS[65] (version 1.16) to visualize macrostructure, especially repeat structure. Contigs were manually assembled into putative models based on their repeat macrostructure. Contig junctions of the putative models were confirmed by aligning genomic reads to them using HiSat2 (version 2.1.0). The large subtelomeric repeat regions flanking the reverse-forward *EPSPS*-cassette duplications were not able to be assembled completely using this method. The locations and relatedness of sequences similar to the 451 bp subtelomeric repeat unit were found using Minimap2 (version 2.24) and BLAST.

### Plot generation

Circos[66] (version 0.69-9) was used to visually summarize the overall *E. indica* genome (Fig. 1). Coverage windows used to make the Circos tracks were generated using bedtools (version 2.30.0; Fig. 1). RIdeogram[67] (version 0.2.2) was used to visualize duplications and deletions of *EPSPS* on chromosome three detected using CNVnator (version 0.4.1; Fig. 2). Synteny plots were made by aligning both genomes using MiniMap2 (version 2.24) and visualized using RIdeogram in R (version 4.2.2; Fig. 3). The *EPSPS*-cassette was visualized using YASS (Fig. 4A, B). Differential expression between eight GS and eight GR *E. indica* individuals was visualized using ggplot2[68] (version 3.4.0) in R. The subtelomere relatedness tree was generated using MUSCLE5 (version 5.1.0), RAXML-NG[69] (version 1.1.0), ggtree[70] (version 3.6.2), cowplot (version 1.1.1; https://cran.r-project.org/web/packages/cowplot/index.html), and ggplot2 in R (Fig. 5).

### Reporting summary

Further information on research design is available in the Nature Portfolio Reporting Summary linked to this article.

## Data availability

The assembled genomes, associated GFF annotation files, and all functional annotation information are publicly available through the International Weed Genomics Consortium online database Weedpedia (https://weedpedia.weedgenomics.org/) under the *Eluesine indica* genome page (https://www.weedgenomics.org/species/eleusine-indica/) or publicly available on CoGe (a platform for performing Comparative Genomics research) under the accession numbers id66361 and id66364, respectively. Genome FASTA files are also available on National Center for Biotechnology Information (NCBI) GenBank under the accessions JARKIM000000000 and JARKIL000000000. Raw sequencing data has been submitted to the NCBI Sequence Read Archive (SRA) numbers: SRR25384219, SRR25384220, and SRR25384221; genome resequencing data with the accession numbers: SRR23364316 - SRR23364331 and RNA-seq data with accession number SRR23372273 - SRR23372288. Source data are provided with this paper, which is also available at Figshare [https://doi.org/10.6084/m9.figshare.23635611][71]. Source data are provided with this paper.

## Code availability

Base code for data analysis and figure generation is available at Github [https://github.com/PattersonWeedLab/Eindica_Subtel_EPSPS_CNV].

Code for genome annotation can also be found at Github [https://github.com/PattersonWeedLab/IWGC_annotation_pipeline].

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

## Acknowledgements

This work is supported by National Science Foundation of China (project nos: 32272568 and 31871984) and the Science foundation of Guangdong Province, China (2019B121201003, 202105TD, R2020PY-JX005) to C.Z. and X.T., and by Michigan State University and the National Science Foundation Research Traineeship Program (DGE-1828149) and a fellowship from Michigan State University under the Training Program in Plant Biotechnology for Health and Sustainability (T32-GM110523) to N.A.J. Funding for Q.Y. by Australian Grains Research and Development Corporation (GRDC) is acknowledged.

## Author contributions

Project was conceived by C.Z., X.T., and Q.Y. Plants were grown, phenotyped, harvested, and submitted for sequencing by C.Z. All sequencing data were analyzed and processed by N.A.J., E.L.P., and N.H. Project hypothesis was generated by N.A.J., E.L.P., and N.H. Final figures and tables were made by N.A.J., E.L.P., and N.H. The manuscript was written by N.A.J., C.Z., N.H., and E.L.P. The manuscript was edited and reviewed by C.Z., N.A.J., N.H., Q.Y., and E.L.P. Major revisions, corrections, and submissions were handled by N.A.J. and E.L.P. Communication and project management was performed by Q.Y., X.T., and E.L.P.

## Competing interests

The authors declare no competing interests.
