## [Peer Review File · Nature Communications]

Subtelomeric 5-enolpyruvylshikimate-3-phosphate synthase copy number variation confers glyphosate resistance in *Eleusine indica*Reviewers' Comments:

Reviewer #1:

Remarks to the Author:

The manuscript by Zhang et., al. assembled and analyzed two genomes of *E. indica*, with one susceptible and another resistant to glyphosate. The authors further used these data to investigate the rearrangement of EPSPS gene into the subtelomeric regions between the two accessions and tried to address their proposed biological question related to the origin and mechanism of the resistant CNVs in weedy species. However, this question is too complex and I do not think that these data and analysis are sufficient to solve the issue. In addition, most of the results are descriptive and some lack validation.

Major issues:

There are some conflicts between results and methods, making their conclusion unreliable.

(1) In Line 95, the authors claimed that the GS genome was sequenced by PacBio HiFi, however, the methods in Line 328 suggest that it was assembled using FALCON with 112.2 Gb Pacbio raw data, which obviously is a pipeline designed for PacBio CLR.

(2) The methods (Lines 390-394) indicated that the re-sequencing was implemented by the Illumina sequencing platform. But results in Line 152 claimed that eight GS and eight GR genomes were re-sequenced using PacBio HiFi rather than Illumina. The author also claimed that they used CNVnator to identify copy number variations based on read alignment generated by HiSat2. Why did not they choose a much more sophisticated genome aligner, such as bowtie or bwa, but use HiSat2 which was designed for RNA-seq mapping?

It is good to know that the authors are fully aware some of their results are not perfect, but I am surprised that they are reluctant to solve those problems. For instance,

(1) Lines 129-131, I understand that assembly of those highly repetitive sequences is difficult, but the recently developed ONT ultra-long sequencing reads can span 100k-200k long regions, which is easily accessible and very likely to resolve those regions.

(2) In Lines 137-140, two large inversions were detected between GR and GS genomes. Given the two genomes were sequenced by two different sequencing approaches and assembled by two different algorithms, the two inversions are likely due to assembly errors. Therefore, validation based on Hi-C data or wet experiments is necessary.

The quality of genome assembly should be extensively evaluated not only by BUSCO. Sup Table 2 shows that the BUSCO scores for the two assemblies reach 98%, however, the completeness of annotation are only 90.5% for GS and 90.4% for GR, which indicates that gene annotation can be refined.

In lines 280-281, the authors claimed that they had strong evidence to support that unequal crossing contributed to the EPSPS CNVs. The conclusion is arbitrary without experimental validation.

Reviewer #2:

Remarks to the Author:

In the manuscript by Zhang et al. the authors uncover the genetic determinants of glyphosate resistance in a monocot species, *Eleusine indica*, a fascinating case of evolution in action. The authors generated high quality reference genomes for both a sensitive and resistant plant and re-sequenced the genomes of a total of 8 resistant and 8 sensitive ones. They could reconstruct the complex structural variants (SVs) associated with resistance.

The manuscript is excellent and remarkably well written, I enjoyed reading it. The experiments are straightforward and well detailed. I was not aware of the work on weed genomes done by this research group but it is quite interesting and relevant in the context of a sustainable agriculture. Glyphosate resistance has been reported in several plant species and requires either several point mutations at EPSPSP or EPSPS overexpression (the target of the herbicide). This overexpression can be conveyed by different mechanisms: gene amplification at genomic loci or at extrachromosomal circles. Here the authors discovered a complex event of EPSPS amplification and rearrangement at

subtelomeric region on chromosome three. Interestingly one plant is heterozygous and present both point mutation and gene amplification suggesting complex dynamics of allele frequency in the population.

The manuscript could be published as it is and will be of interest to Nat Comm readers working on genome evolution, herbicide resistance and SVs, and beyond. The resource created might also be helpful for plant genomics.

If to suggest very minor points:

Line 45 a reference for the influence of the epigenome on SVs could be helpful.

Line 51-52: same comment

Lines 110 & 116 : Gypsy and Copia are generally italicized (also in Figures)

Line 331 maybe precise 30x coverage (as stated later)

Line 649: "Gene labels with a non-integer numerical value represent splice variants of the same gene." Not sure it applies here.

Reviewer #3:

Remarks to the Author:

The research presented in the manuscript is novel and adds significantly to the genomics aspects of herbicide resistance evolution. However, the manuscript needs significant work. Palmer amaranth needs to be written with "P" in caps and also full name "Palmer amaranth" It is not scientific to mention the weeds as "palmer" (line 77 and 79).

Also, there is no mention of the EPSPS copy number of glyphosate-resistant and susceptible plants in the results section, although in materials and methods section that was mentioned.

At the end of results and discussion section, the authors suggest that they have strong evidence that the EPSPS CNV is due to unequal crossover with initial translocation event at sub-telemeric region, however in the conclusion (line 291-292) they say, they still hypothesize about this finding. This is confusing

Important reference that involves the mechanism of EPSPS amplification in weed species were missing, especially regarding unequal crossover.. literature review was poorly done, and the formatting of literature is not uniform throughout.

It is important to pay attention to the details in the methods section as well. On page 7. line 315, it is given that EPSPS CVN was assessed from resistant population, this should be from the resistant plants not "population".

Also, in methods line 302 to 303, not sure what authors mean by purified.. this should be revised for clarity

Reviewer #4:

Remarks to the Author:

Eric Patterson and his colleagues sequenced two high-quality genomes (herbicide lyphosate-susceptible and -resistant lines) of Eleusine indica, an important weed. They found that (EPSPS) copy number variation confers the resistance by a novel genomic arrangement with subtelomeric repeats. Further lyphosate-susceptible and -resistant population (8 lines per population) re-sequencing and transcriptomic investigation confirmed the finding. The study provides important reference genomes for the weedy species and adds new knowledge about herbicide resistance by large-scale

genomic mechanism.

Minor comments:

- 1.The title of manuscript failed to cover their genome efforts, i.e. their main goal is to sequence genome of *E. indica*. It will be better to add "genome" or like word in their title.
- 2.Fig. 1 and Fig. 5 provide less information and should be move as supplementary figures.
- 3.It will be better to provide a phylogeny tree of *E. indica* in the manuscript. As an important weed species to be sequenced, how many other important weeds were sequenced to chromosome-level?

RESPONSE TO REVIEWER COMMENTS

Reviewer #1:

The manuscript by Zhang et., al. assembled and analyzed two genomes of *E. indica*, with one susceptible and another resistant to glyphosate. The authors further used these data to investigate the rearrangement of EPSPS gene into the subtelomeric regions between the two accessions and tried to address their proposed biological question related to the origin and mechanism of the resistant CNVs in weedy species. However, this question is too complex and I do not think that these data and analysis are sufficient to solve the issue. In addition, most of the results are descriptive and some lack validation.

Answer: We understand your point; however, we believe as an initial study, our work represents the first steps in addressing this daunting task. Indeed, with more research and time much can be revealed in the near future.

Major issues:

There are some conflicts between results and methods, making their conclusion unreliable.

(1) In Line 95, the authors claimed that the GS genome was sequenced by PacBio HiFi, however, the methods in Line 328 suggest that it was assembled using FALCON with 112.2 Gb Pacbio raw data, which obviously is a pipeline designed for PacBio CLR.

Answer: Thank you for noticing this. You are correct. This was a typo which we have revised. PacBio HiFi was used for sequencing the GR genome only.

(2) The methods (Lines 390-394) indicated that the re-sequencing was implemented by the Illumina sequencing platform. But results in Line 152 claimed that eight GS and eight GR genomes were re-sequenced using PacBio HiFi rather than Illumina.

Answer: Thank you very much for catching this. The re-sequencing was indeed completed using Illumina HiSeq X Ten. We have updated the results to address this. We have thoroughly combed through the manuscript for any similar typos.

The author also claimed that they used CNVnator to identify copy number variations based on read alignment generated by HiSat2. Why did not they choose a much more sophisticated genome aligner, such as bowtie or bwa, but use HiSat2 which was designed for RNA-seq mapping?

Answer: Thank you for your suggestion. However, we argue that HiSat2 is a sophisticated alignment tool that can align reads to both genomics and transcriptomic references. The HiSat2 manual page specifically states HiSat2 was designed for use with both genomic and transcriptomic alignment. "HISAT2 is a fast and sensitive alignment program for mapping next-generation sequencing reads (whole-genome, transcriptome, and exome sequencing data)" (<http://daehwankimlab.github.io/hisat2/manual/>). Furthermore, HiSat2 was developed of the same base code as Bowtie but has been refined for speed and accuracy (there are other benefits as well.) BWA is a great choice, in fact we used it initially; however, it was far slower mapping the number of reads we had. We ran several samples with BWA and Hisat2 and ultimately it made no difference on the CNVnator analysis.

(1) It is good to know that the authors are fully aware some of their results are not perfect, but I am surprised that they are reluctant to solve those problems. For instance, Lines 129-131, I understand that assembly of those highly repetitive sequences is difficult, but the recently developed ONT ultra-

long sequencing reads can span 100k-200k long regions, which is easily accessible and very likely to resolve those regions.

Answer: Thank you for your suggestion for ONT. Although ONT may be able to span some of these repeat regions, the EPSPS-Cassette itself is around 100kb in length and flanked by highly repetitive regions, at a minimum of 50Kb each. We are unsure if even ultra-long ONT would be able to resolve these regions. We believe that either BioNano or fiber-FISH could resolve these regions but both are outside of our capabilities and budget.

(2) In Lines 137-140, two large inversions were detected between GR and GS genomes. Given the two genomes were sequenced by two different sequencing approaches and assembled by two different algorithms, the two inversions are likely due to assembly errors. Therefore, validation based on Hi-C data or wet experiments is necessary.

Answer: Thank you for bringing up this valid point. The inversions could potentially be an artifact of the specific algorithms used for the two assemblies. To validate the inversions, we mapped resistant reads to the inversion junctions from both the GR and GS genome assemblies. For the CHR5 inversion, many reads support the inversion of the arm of CHR5 in the GR genome while no reads support this locus in the GS orientation. We believe that this is strong evidence that the inversion in CHR5 in the GR gene is true. On the other hand, confirming the smaller, internal inversion of CHR3 is more challenging. Due to the presence of highly repetitive DNA flanking this locus, reads mapped poorly to this particular region. A few reads did support the inversion, and none support the GS orientation, however, we lack complete confidence. Therefore, this may or may not be a true inversion distinguishing the lines. We have made two new supplemental figures (Supp. Figs 2 and 3) that show the read mapping and have updated the text to explain this analysis in Methods and Results.

The quality of genome assembly should be extensively evaluated not only by BUSCO. Sup Table 2 shows that the BUSCO scores for the two assemblies reach 98%, however, the completeness of annotation are only 90.5% for GS and 90.4% for GR, which indicates that gene annotation can be refined.

Answer: We appreciate your insightful comments. We would like to note that Supp. Table 1 and 2 features assembly metrics in addition to the BUSCO scores. Generally speaking, when running BUSCO on annotated genes, the scores tend to be lower than those on genomes. This is due to many reasons including missed genes in the annotation. Additionally, BUSCO finds fragmented and/or pseudogenized genes in the genome that are not expressed (or at least not in normal tissues/circumstances used for annotation). For instance, BUSCO using the embryophyta_odb10 database on the most recent rice genome reports a “completeness” of 98.2% while the completeness score is 93.0% when considering only annotated genes. We did, however, rerun BUSCO using the most recent embryophyta_odb10 database since it was released after we began this project. We have updated our BUSCO scores, which are on par with rice:

Glyphosate-Susceptible	Final assembly	97.8%
Glyphosate-Susceptible	Genome Annotation	92.1%
Glyphosate-Resistant	HiCanu assembly	97.8%
Glyphosate-Resistant	Genome Annotation	92.2%

We also performed a LTR Assembly Index (LAI) analysis for measuring assembly completeness of our GS and GR assemblies. The results have been added to supplemental table 2. The LAI score of GS is 18.77, while for GR is 16.85. According to the paper: Ou, Shujun, Jinfeng Chen,

and Ning Jiang. "Assessing genome assembly quality using the LTR Assembly Index (LAI)." *Nucleic acids research* 46, no. 21 (2018): e126-e126, a score between 10-20 can be considered 'reference grade' assemblies, while greater than 20 is "gold", so while our assemblies may still fall below the criteria for gold assembly, they are still valuable reference assemblies for weed scientists working on this and other grass species.

In lines 280-281, the authors claimed that they had strong evidence to support that unequal crossing contributed to the EPSPS CNVs. The conclusion is arbitrary without experimental validation.

Answer: Thank you and this is a very valid point as we do not directly measure crossing over. We have changed our wording significantly in the results to indicate that this is only a hypothesis and that we do not have direct evidence of crossing over.

Reviewer #2 (Remarks to the Author):

In the manuscript by Zhang et al. the authors uncover the genetic determinants of glyphosate resistance in a monocot species, *Eleusine indica*, a fascinating case of evolution in action. The authors generated high quality reference genomes for both a sensitive and resistant plant and re-sequenced the genomes of a total of 8 resistant and 8 sensitive ones. They could reconstruct the complex structural variants (SVs) associated with resistance.

The manuscript is excellent and remarkably well written, I enjoyed reading it. The experiments are straightforward and well detailed. I was not aware of the work on weed genomes done by this research group but it is quite interesting and relevant in the context of a sustainable agriculture.

Answer: We are very happy you enjoyed reading our manuscript! Thank you for your kind words.

Glyphosate resistance has been reported in several plant species and requires either several point mutations at EPSPSP or EPSPS overexpression (the target of the herbicide). This overexpression can be conveyed by different mechanisms: gene amplification at genomic loci or at extrachromosomal circles. Here the authors discovered a complex event of EPSPS amplification and rearrangement at subtelomeric region on chromosome three. Interestingly one plant is heterozygous and present both point mutation and gene amplification suggesting complex dynamics of allele frequency in the population.

The manuscript could be published as it is and will be of interest to Nat Comm readers working on genome evolution, herbicide resistance and SVs, and beyond. The resource created might also be helpful for plant genomics.

If to suggest very minor points:

Line 45 a reference for the influence of the epigenome on SVs could be helpful.

Answer: Thank you for this suggestion, we have added the suggested reference.

Line 51-52: same comment

Answer: Thank you again. The reference has been updated.

Lines 110 & 116 : Gypsy and Copia are generally italicized (also in Figures)

Answer: Thank you for catching this typo. We have italicized these names.

Line 331 maybe precise 30x coverage (as stated later)

Answer: Thank you for this suggestion. We have revised the text as you suggested.

Line 649: "Gene labels with a non-integer numerical value represent splice variants of the same gene." Not sure it applies here.

Answer: Thank you for catching this error. This was a remnant of an old header which has been removed.

Reviewer #3 (Remarks to the Author):

The research presented in the manuscript is novel and adds significantly to the genomics aspects of herbicide resistance evolution. However, the manuscript needs significant work. Palmer amaranth needs to be written with "P" in caps and also full name "Palmer amaranth" It is not scientific to mention the weeds as "palmer" (line 77 and 79).

Answer: Thank you for the positive comments and for pointing these typos out. We have updated these instances to *Amaranthus palmeri*. The manuscript has been significantly improved following reviewers' comments/suggestions.

Also, there is no mention of the EPSPS copy number of glyphosate-resistant and susceptible plants in the results section, although in materials and methods section that was mentioned.

Answer: Thank you for your comment. Lines 172-179 in subsection "2.2 EPSPS Structural Variation" of the Results and Discussion section discusses EPSPS CNV in GR and GS. Specifically, we say that:

"Seven of the eight GR individuals had EPSPS read depths of approximately 25-29x above background with the exception being sample R14 which only had 8x (Fig. 3)."

We feel this portion of the results addresses your comment. Resequencing of Susceptible plants show no increased read depth at the EPSPS loci and therefore are at one copy (See figure 3).

We have now added the below sentence for more clarity that these are resistance specific CNVs.

"All GS individuals had normal read depths for Region-A and -B and therefore no CNV in either Region-A or -B (Fig. 3)."

At the end of results and discussion section, the authors suggest that they have strong evidence that the EPSPS CNV is due to unequal crossover with initial translocation event at sub-telemeric region, however in the conclusion (line 291-292) they say, they still hypothesize about this finding. This is confusing.

Answer: Thank you for pointing this out and is an excellent point, which aligns with the feedback by Reviewer 1. This is indeed a hypothesis based on past literature about subtelomeric gene amplification and it is implied by our findings. However, our work is not conclusive on whether or not crossing over is the main or only mechanism of EPSPS CNV formation. In response to the comments, we have revised the language in the last paragraph of the results, removing the phrase “strong evidence for crossing over”. Instead, we now present it as one possible mechanism for duplication.

Important reference that involves the mechanism of EPSPS amplification in weed species were missing, especially regarding unequal crossover..

Answer: We appreciate this feedback. As far as we are aware, EPSPS CNV has not been proven to be caused by unequal crossover; however, it has been strongly hypothesized in two papers in the case of Tandem duplication of EPSPS in *Kochia scoparia*:

Jugulam, Mithila, Kindsey Niehues, Amar S. Godar, Dal-Hoe Koo, Tatiana Danilova, Bernd Friebe, Sunish Sehgal et al. "Tandem amplification of a chromosomal segment harboring 5-enolpyruvylshikimate-3-phosphate synthase locus confers glyphosate resistance in *Kochia scoparia*." *Plant physiology* 166, no. 3 (2014): 1200-1207

Patterson, Eric L., Christopher A. Saski, Daniel B. Sloan, Patrick J. Tranel, Philip Westra, and Todd A. Gaines. "The draft genome of *Kochia scoparia* and the mechanism of glyphosate resistance via transposon-mediated EPSPS tandem gene duplication." *Genome biology and evolution* 11, no. 10 (2019): 2927-2940.

We failed to cite the paper by Jugulam et al (2014) and have now corrected this.

Additionally, similar adaptive evolution for other stress has been observed to be caused by tandem duplications in the subtelomeres, with the duplications caused by unequal crossover. Our citations in the results and discussion section 2.4 *Subtelomeres in plant evolution* reference these findings.

literature review was poorly done, and the formatting of literature is not uniform throughout.

Answer: Thank you for this feedback, however, we would like some clarification regarding the comment on the literature review being "poorly done.". Is there specific literature we should have included? We had already reached the 70-citation limit that Nature Communications has set. We would be happy to further revise the literature review based on your suggestions. We did however, go through the literature cited and curate the citations to have as much detail as possible.

It is important to pay attention to the details in the methods section as well. On page 7. line 315, it is given that EPSPS CNV was assessed from resistant population, this should be from the resistant plants not "population".

Answer: Thank you for the fair comment. We have refined our language as suggested.

Also, in methods line 302 to 303, not sure what authors mean by purified. this should be revised for clarity.

Answer: Thank you for this comment. We have changed this line to read term “purified” to instead say “Well phenotyped individuals (i.e. confirmed susceptible and resistant) were self-pollinated” for increased homozygosity as this was our meaning of the word “purified”.

Reviewer #4 (Remarks to the Author):

Eric Patterson and his colleagues sequenced two high-quality genomes (herbicide glyphosate-susceptible and -resistant lines) of *Eleusine indica*, an important weed. They found that (EPSPS) copy number variation confers the resistance by a novel genomic arrangement with subtelomeric repeats. Further glyphosate-susceptible and -resistant population (8 lines per population) re-sequencing and transcriptomic investigation confirmed the finding. The study provides important reference genomes for the weedy species and adds new knowledge about herbicide resistance by large-scale genomic mechanism.

Answer: Thank you for the positive comments.

Minor comments:

1. The title of manuscript failed to cover their genome efforts, i.e. their main goal is to sequence genome of *E. indica*. It will be better to add “genome” or like word in their title.

Answer: Thank you for this very good suggestion and we agree. We have changed the title to “*Chromosome-level assemblies reveal subtelomeric 5-enolpyruvylshikimate-3-phosphate synthase (EPSPS) copy number variation confers glyphosate resistance in *Eleusine indica**”, to reflect the significance of the genomic data produced.

2. Fig. 1 and Fig. 5 provide less information and should be move as supplementary figures.

Answer: Thank you for this suggestion. We agree that Fig. 5 was simply to confirm that EPSPS was overexpressed, not just duplicated. Fig. 5 also shows genes on the EPSPS-Cassette which are differentially expressed. We can move Fig. 5 to supplementary as supplementary Figure 4 in the revised version. For Fig. 1, we acknowledge that our study primarily focuses on the EPSPS locus rather than being a comprehensive whole-genome analysis. As you commented above about the significance of producing a reference genome for this important weed as a notable accomplishment, we believe that including Fig. 1 in the main manuscript would provide readers with essential genomic orientation and context for the overall narrative.

3. It will be better to provide a phylogeny tree of *E. indica* in the manuscript. As an important weed species to be sequenced, how many other important weeds were sequenced to chromosome-level?

Answer: Thank you again for suggesting a phylogeny. We did perform some initial phylogenetic work for comparison genomics; however, our story became more focused on glyphosate resistance and less on analyzing the genome or goosegrasses macroevolution. We feel that adding phylogenetic analysis to this paper would dilute the central story, however, it would be valuable in a follow up story about a broader grass weed genome evolution, especially as other species are now being published.

Reviewers' Comments:

Reviewer #1:

Remarks to the Author:

I appreciate the authors' efforts to improve the manuscript. While there may still be some room for improvement in the results (e.g., sequencing ONT ultra-long reads to improve genome assembly), they have adequately addressed all of my major concerns. Therefore, I have no further comments to add at this time.

Reviewer #2:

Remarks to the Author:

Thank you to the authors for their clear answers. All my comments have been addressed. I only list below a few minor points that could help.

Concerning the reference 7 I think it relates to nuclear architecture (3D genome) rather than epigenome, this could be clarified in the text. I don't think there are many studies on the impact of the epigenome on SVs (at least in plants).

Please note a typo at line 459 for the minimap2 reference (ref 53).

Regarding one remark made by Reviewer#3 I noticed that Palmer amaranth was not edited as suggested (capital P missing, line 87). It was named after the botanist Edward Palmer (hence the capital letter), see <https://doi.org/10.2307/2394403>.

Reviewer #4:

Remarks to the Author:

No further comments

REVIEWERS' COMMENTS

Reviewer #1 (Remarks to the Author):

I appreciate the authors' efforts to improve the manuscript. While there may still be some room for improvement in the results (e.g., sequencing ONT ultra-long reads to improve genome assembly), they have adequately addressed all of my major concerns. Therefore, I have no further comments to add at this time.

Thank you for your help in review. We appreciate all of the thoughtful and helpful edits.

Reviewer #2 (Remarks to the Author):

Thank you to the authors for their clear answers. All my comments have been addressed. I only list below a few minor points that could help.

Concerning the reference 7 I think it relates to nuclear architecture (3D genome) rather than epigenome, this could be clarified in the text. I don't think there are many studies on the impact of the epigenome on SVs (at least in plants).

That was a mis-citation, we have changed the reference to no longer be attributed to epigenetics.

Please note a typo at line 459 for the minimap2 reference (ref 53).

We have fixed this typo.

Regarding one remark made by Reviewer#3 I noticed that Palmer amaranth was not edited as suggested (capital P missing, line 87). It was named after the botanist Edward Palmer (hence the capital letter), see <https://doi.org/10.2307/2394403>.

We have fixed this capitalization error.

Reviewer #4 (Remarks to the Author):

No further comments